# Uniportal Video-Assisted Thoracoscopic Segmentectomy for Early-Stage Non-Small Cell Lung Cancer: Overview, Indications, and Techniques

**DOI:** 10.3390/cancers16132343

**Published:** 2024-06-26

**Authors:** Takuya Watanabe, Masayuki Tanahashi, Eriko Suzuki, Naoko Yoshii, Takuya Kohama, Kensuke Iguchi, Takumi Endo

**Affiliations:** Division of Thoracic Surgery, Respiratory Disease Center, Seirei Mikatahara General Hospital, 3453, Mikatahara-cho, Chuo-ku, Hamamatsu 433-8558, Japan; m.tanahashi@sis.seirei.or.jp (M.T.); e-suzuki@sis.seirei.or.jp (E.S.); n-yoshii@sis.seirei.or.jp (N.Y.); 131232.tk@gmail.com (T.K.); iguchik47@gmail.com (K.I.); endo.t@sis.seirei.or.jp (T.E.)

**Keywords:** uniportal video-assisted thoracoscopic surgery, thoracic surgery, VATS, uniportal VATS, U-VATS, segmentectomy, single-port, early-stage, non-small cell lung cancer

## Abstract

**Simple Summary:**

Uniportal video-assisted thoracoscopic segmentectomy is one of the most minimally invasive surgeries for early-stage lung cancer. However, mastering this technique requires high levels of skill and experience, especially in complex cases. This article reviews previous research on uniportal video-assisted thoracoscopic segmentectomy, including indications, instrument selection, tumor marking, intersegmental plane identification, lymph node dissection, and surgical videos.

**Abstract:**

Twenty years have passed since uniportal video-assisted thoracoscopic surgery (VATS) was first reported. Several reports have already proven the minimal invasiveness of uniportal VATS. In addition, two large clinical trials recently demonstrated the benefits of segmentectomy for small peripheral early-stage non-small cell lung cancer. Uniportal VATS segmentectomy is considered the most beneficial minimally invasive surgery for patients with early-stage lung cancer. However, a high level of skill and experience are required to achieve this goal. Only a few reports have discussed specific techniques, particularly for complex segmentectomies. In this Special Issue, we reviewed previous reports on uniportal VATS segmentectomy regarding the indications, instrument selection, marking of the tumor location, methods of intersegmental plane identification, and lymph node dissection, including our own techniques with video content.

## 1. Introduction

In 2004, the first lung resection using uniportal video-assisted thoracoscopic surgery (U-VATS) was reported [1]. U-VATS lobectomy was first reported in 2012 [2]. In the years that followed the first report, this approach has spread worldwide [3,4]. Uniport surgeons have continued to develop their techniques [5,6]. The indications for their surgeries have been expanded to include highly difficult surgeries, such as segmentectomy [7], inflammatory lymph node cases [8], bronchoplasty [9], and angioplasty [10]. U-VATS has many advantages over conventional multiportal VATS (M-VATS), which uses 3–5 ports, including less postoperative pain and complications, shorter operative time, and shorter hospital stay [11,12,13,14,15,16,17,18,19,20,21].

The benefit of segmentectomy for early-stage peripheral non-small cell lung cancer (NSCLC) has recently been demonstrated in large clinical trials [22,23]. Against this background, segmentectomy with U-VATS is considered the most beneficial minimally invasive surgery for patients with early-stage NSCLC. However, a high level of skill and experience are required to perform the operation [24,25,26,27]. In particular, few reports have discussed specific techniques, including U-VATS complex segmentectomy.

In this Special Issue, we review previous reports on U-VATS segmentectomy, including studies discussing the indications, instrument selection, marking of the tumor location, methods of intersegmental plane identification, and lymph node dissection. We also discuss our own techniques.

## 2. About Uniportal VATS

### 2.1. Definition of Uniportal VATS

At the consensus meeting of the Uniportal VATS Interest Group (UVIG) of the European Society of Thoracic Surgeons (ESTS), U-VATS was defined as “VATS performed from a single incision of 4 cm or less” [24]. In this article, U-VATS is described based on this definition.

### 2.2. Comparison with Multiportal VATS

Several studies comparing perioperative outcomes between U-VATS and M-VATS are shown in Table 1. Each report demonstrated the superiority or equivalence of U-VATS for almost all of these parameters, even in patients of ≥70 years of age [12]. Three systematic reviews and meta-analyses have concluded that U-VATS is superior to traditional thoracotomy in terms of postoperative hospital stay, postoperative drainage duration, complication rate, conversion rate to thoracotomy, postoperative pain, and need for analgesics [28,29,30]. There are also reports of reduced medical costs due to faster postoperative recovery [31] and a reduced incidence of long-term post-thoracotomy pain syndrome [14]. In terms of prognosis, the 3-year overall survival (OS) rates of U-VATS and M-VATS in lobectomy were 74% and 76%, respectively, the 3-year disease-free survival (DFS) rates were 69% and 67% (including advanced lung cancer) [12], the 5-year OS rates were 89.2% and 86.5%, and the 5-year DFS rates were 89.5% and 89.6% [11]. In segmentectomy, the 3-year OS rates of U-VATS and M-VATS were 100% and 90.4%, respectively, the 3-year DFS rates were 99.7% and 99.0%, the 5-year OS rates were 97.7% and 99.4%, and the 5-year DFS rates were 99.7% and 98.2% [18]. There were no significant differences between the two groups [11,12,18]. Although these are mostly retrospective studies, almost all reports have demonstrated the superiority of U-VATS over M-VATS, and more than half (58%) of the participants at the UVIG meeting of the ESTS agreed that a randomized controlled trial comparing U-VATS and M-VATS is unnecessary [24].

## 3. Before Performing Uniportal VATS Segmentectomy

### 3.1. Required Skills and Experience

U-VATS is more difficult than M-VATS because it is prone to surgical instrument interference. In addition, U-VATS requires the ability to control and perform surgeries independently because it is generally difficult to obtain help from assistants. In the UVIG report of ESTS, 84% of the members agreed that more than 50 cases were needed to reach the learning curve for lobectomy in U-VATS [24]. A study analyzing 1063 cases and examining the learning curve reported that the operative time plateaued at 40 U-VATS cases [27]. A study limited to U-VATS segmentectomy reported that it took 70 cases to reach the learning curve and 100 cases to achieve proficiency [26]. In contrast, U-VATS segmentectomy can be safely performed by surgeons with extensive experience in U-VATS [25], and U-VATS complex segmentectomy can be performed without any learning curve by surgeons who are experienced in U-VATS lobectomy or simple segmentectomy techniques [32]. With this background, it can be assumed that U-VATS simple or complex segmentectomy can be performed at a certain level with basic techniques and experience in U-VATS lobectomy.

### 3.2. Indications for Uniportal VATS Segmentectomy

Based on the results of JCOG0802/WJOG4607L [22] and CALGB 140503 [23], early-stage peripheral NSCLC of less than 2 cm in size is a good indication for segmentectomy. The number of cases is expected to increase with the spread of lung cancer screening using computed tomography (CT) [33,34,35]. However, palpation is often difficult with U-VATS, and in cases where tumor localization cannot be confirmed from the pleura, preoperative marking, as described below, should be used. We do not consider U-VATS to be a good approach for cases that are somewhat deeper in the lung parenchyma that should be resected by the palpation of the tumor, especially common in metastatic lung tumors. U-VATS segmentectomy is a good approach for minimizing the impact on the systemic condition in elderly patients with a poor performance status and patients with a low pulmonary function who cannot undergo wedge resection. At present, there is no evidence to support segmentectomy for NSCLC with lymph node metastasis, and in principle, it should only be performed in patients with clinical N0 disease, except for the passive indications mentioned above.

### 3.3. Preoperative Planning

In segmentectomy, it is important to (1) localize the tumor, (2) ensure surgical margins, and (3) understand the detailed anatomy.

#### 3.3.1. Locating the Tumor

Since palpation is often difficult in U-VATS, it is important to visualize the localization of the tumor by preoperative marking. These methods include CT-guided hook wire [36], spiral wire [37], butterfly needle [38], microcoil [39], dye [40], lipiodol [41], electromagnetic navigation bronchoscopy [42], and intraoperative ultrasonography [43]. A novel application of the radiofrequency identification marking system was recently reported [44,45,46,47]. It has also been reported that the combined use of mechanical and chemical markings reduces the rate of conversion to thoracotomy in VATS. The success rate of the hook-wire has been reported to be 98.3% [48], and many institutions have made it their first choice [36,48,49,50,51]. However, several cases of air embolization, including fatal cases, have been reported [52,53,54,55], and hook wire marking is not recommended in Japan. A report of preoperative dye marking in U-VATS showed a 99.5% success rate of implementation and a 100% success rate of tumor resection without conversion to thoracotomy [56]. Similar to this report, our institution uses CT-guided dye marking and has had no experience of fatal complications. We perform preoperative marking in cases where the tumor may not be visible from the pleura and where the lesion mainly demonstrates ground-glass opacity. In addition, we consider that cases in which the tumor is located somewhat deep in the lung and thus may move in the parenchyma are not good indications for U-VATS segmentectomy, and such cases should instead be dissected between the intersegmental planes with palpation under thoracotomy.

#### 3.3.2. Ensuring Tumor Margins

For solid-predominant lesions, the tumor margin should be 2 cm or larger than the tumor diameter [22]. For ground-glass opacity-predominant lesions, the tumor margin should be 5 mm or more [57,58]. The post hoc analysis of JCOG0802 showed the benefit of segmentectomy over lobectomy, even in pure-solid tumors [59], and it would be acceptable to ensure the above-mentioned tumor margin. In addition, because complex segmentectomy has been reported to have equivalent outcomes to simple segmentectomy [60], performing a combined resection of adjacent segments to ensure the tumor margin should also be considered.

#### 3.3.3. Anatomical Variations

The 3D reconstruction of pulmonary arteries (PAs), pulmonary veins (PVs), and bronchi using an imaging analysis software program is highly recommended for segmentectomy [61,62,63]. At our institution, all patients underwent 3D reconstruction using Synapse Vincent (ver. 6.7, Fujifilm, Tokyo, Japan). Contrast-enhanced CT is the standard. However, plain CT can also be used to obtain a rough image of the vascular branching. Based on this information, the segments and vessels and bronchi to be resected can be thoroughly reviewed before surgery. For example, it has been reported that the anatomy of the PVs in the right upper lobe is extremely variable [64], and changes in the surgical procedure due to anatomical abnormalities should also be considered.

## 4. Surgical Techniques of Uniportal VATS

### 4.1. Patient Position

A patient is placed in the lateral recumbent position, and the surgical bed is tilted approximately 15 degrees at the top of the intercostal space (ICS) to be used for a uniport (Figure 1). This allows the ICS to open and is useful in cases of conversion to open thoracotomy. The hand on the affected side is placed on a handstand. In the event of emergency conversion to thoracotomy, a new axillary incision should be added at the 4th ICS to provide a proper view of the pulmonary hilum. If a uniport is not created at the 4th ICS, the first uniport should be used as an assist or camera port.

### 4.2. Approach and Incision

While many uniport surgeons use the anterior approach, standing on the ventral side of the patient [49,51,61,65,66], the first author always stands on the right side of the patient (Figure 2). This maximizes the use of the dominant hand for the manipulation of energy devices and staplers. The first assistant and scopist stand on the opposite side of the main surgeon. A uniport is created in the mid-axillary line for right-sided surgery and in the anterior axillary line for left-sided surgery. The strength of this approach is that it provides a bird’s-eye view of the thoracic cavity. It is, therefore, possible to secure a view of not only the hilum but also the dorsal side, thus making it suitable for dorsal pleural dissection and lymph node dissection in the upper and lower mediastinal regions. However, the ICS is slightly narrower at the mid-axillary line than at the anterior axillary line, which may be a disadvantage in small patients with a narrow ICS.

The ICS for uniport placement is determined by visualizing the location of the pulmonary hilum and interlobe in the thoracic cavity, without counting the ribs from the body surface. If there is any doubt regarding the ICS between the upper and lower ribs, it is better to select the lower ICS because manipulation is extremely difficult if the uniport is even slightly cephalad to the target pulmonary hilum.

**Figure 1 cancers-16-02343-f001:**
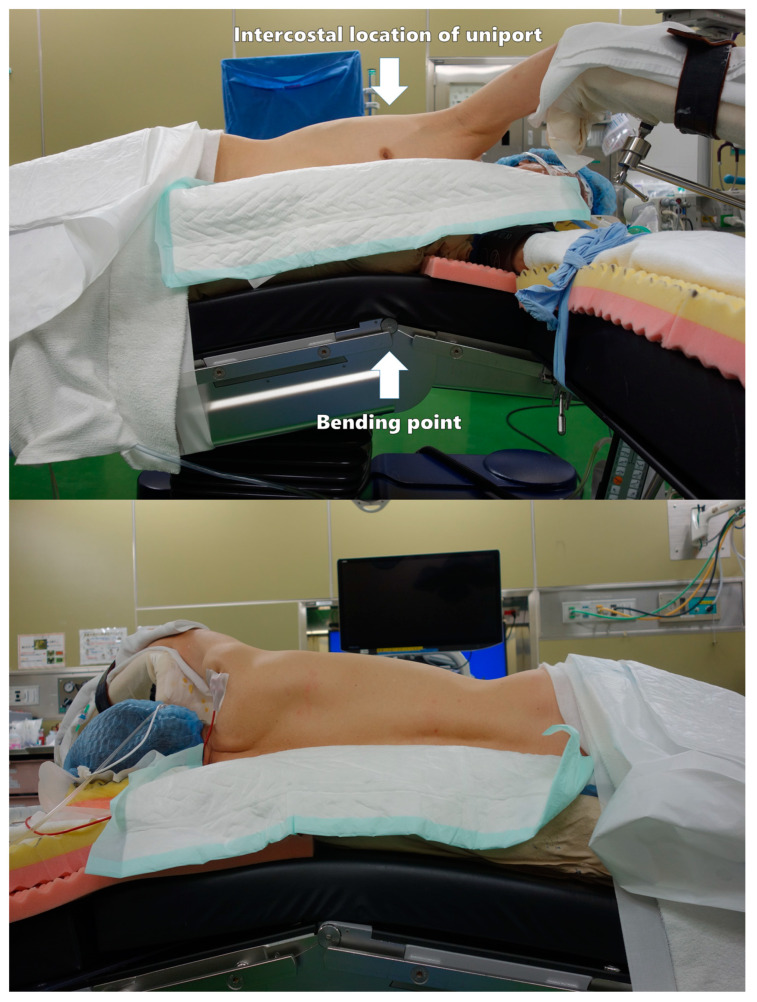
Patient position.

**Figure 2 cancers-16-02343-f002:**
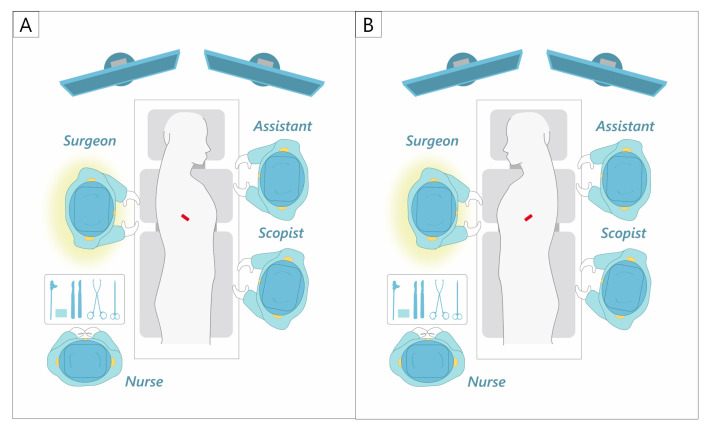
Our uniportal VATS approach and setting. (**A**) Right side. (**B**) Left side.

A polyurethane retractor is placed on the wound to protect the intercostal space. A 14-Fr suction catheter is inserted through the edge of the retractor as a smoke evacuation device (Figure 3). This method has been reported to significantly reduce intraoperative camera clearing [67]. As the suction device does not need to be placed in the thoracic cavity, the left hand can be tubeless.

### 4.3. Surgical Instruments of Uniportal VATS

Several curved forceps need to be prepared, including suction tools, lung forceps, lymph node forceps, right-angle forceps, Kelly forceps, snake forceps, and curved forceps (Figure 4A). The thoracoscope is ENDOEYE (Olympus, Tokyo, Japan) with a 5 mm 30-degree rigid integrated scope. Energy devices are essential in U-VATS, although surgeons can choose their own devices according to their preferences. The first author frequently uses a 1 mm wide hook-type electrocautery scalpel, AdTec^®^ mini (B.BRAUN AESCULAP, Melsungen, Germany) (Figure 4B) for the incision of vessels and bronchial sheaths. A 23 cm curved suction soft coagulator (AMCO, Tokyo, Japan) (Figure 4C) is also useful for dissection and hemostasis. The left hand is mainly used with the CS Two-Way HandleTM (Unimedic, Osaka, Japan) (Figure 4D). It is also useful for compression hemostasis in cases with oozing [68]. They are characterized by greater friction than suction instruments, making it easier to adjust the tension. Techniques using these instruments are presented with surgical videos in the segmentectomy case series section.

### 4.4. Vascular and Bronchial Dissection

In U-VATS, the main method of dissection is the opening of the energy device; however, the first author mainly uses hook-type electrocautery (Figure 4B). A sharp incision of the membrane facilitates anatomical understanding and a dry surgical field. The PA, PV, and bronchi should be cut with reference to preoperative 3D reconstruction after confirming that the orientation is correct. When dissecting the PA or PV, the stapler is inserted by sliding it over the Penrose drain. Two rows of staplers can also be used for (sub) segmental bronchi in U-VATS segmentectomy.

### 4.5. Lymph Node Dissection

Owing to the limited number of tools available for U-VATS, lymph node dissection (LND) needs to be designed. In a sub-analysis of JCOG0802, the frequency of upstaging (N2) by LND was 0.5% in part-solid tumors, whereas N2 was identified in 5.4% of pure-solid tumors. Therefore, selective LND is recommended [69,70]. There are also reports of a good prognosis in cases where more than 10 lymph nodes were dissected in segmentectomy [71], and in U-VATS segmentectomy, LND should be performed appropriately according to the case. For the dissection of the N1 lymph nodes, a sharp incision should be made through the vascular or bronchial sheath, and the lymph nodes should be dissected under the surrounding capsule; hook electrocautery is useful for sharp incisions in U-VATS (Video 1). The careful dissection of the N1 lymph nodes without crushing is particularly important to ensure curability in U-VATS segmentectomy for early-stage NSCLC. For superior mediastinal LND, dissection around the azygos vein before dissecting the superior trunk (A3) eliminates the need for an assistant to protect the A3 stump. The azygos vein can be taped and towed by an assistant to provide a good surgical field for LND (Figure 5A). For inferior mediastinal LND, a good surgical field around the tracheal bifurcation can be obtained by passing tape between the superior and inferior PVs and towing the entire lower lobe forward (Figure 5B). For this reason, inferior mediastinal LND is performed before cutting the inferior PV and inferior lobe bronchus during lower lobectomy.

### 4.6. Identification of the Intersegmental Plane

Various methods have been used to identify intersegmental planes, including the inflation–deflation technique [72], jet ventilation [73], and intravenous indocyanine green (ICG) injection [74]. ICG injection is considered useful because of the limitations of the surgical field and the use of instruments in U-VATS [75]. Our institution uses ICG. It has also been reported that ICG injection using a syringe pump significantly clarifies the contrast between the circulation area and the ischemic area in comparison to bolus injection [76]. Specifically, 25 mg of ICG is dissolved in 10 mL of saline, with a maximum dose of 0.3 mg/kg, and administered intravenously at a rate of 300 mL/h. The contrast is visualized in approximately 30 s, and if the maximum dose is not reached, the administration is stopped. After 30 min, the ICG is washed out and the procedure can be repeated. The intersegmental plane should be adequately marked with dye.

### 4.7. Stapling between Intersegmental Planes

This is the most important step in U-VATS segmentectomies. Specifically, in cases where the tumor is not visible from the pleura, it is necessary to depend on preoperative markings to determine the appropriate staple line to ensure the tumor margin. All peripheral stumps of the dissected intersegmental PV or (sub) segmental bronchus are ligated and towed. The control of these threads facilitates stapling between the intersegmental planes (Figure 6). After stapling the intersegmental planes, the specimen should be thoroughly palpated. The staples should be removed and bled off to facilitate palpation. If no tumor is palpable in the specimen, the surgeon should not hesitate to convert to thoracotomy and palpate the lung again, even if the surgery is almost complete.

### 4.8. Before the End of Surgery

A sealing test is performed to confirm the absence of bronchial and pulmonary fistula. If an air leak is identified, it should be carefully repaired because it significantly affects the postoperative course. If the air leak is mild, it can be controlled with soft coagulation, and the use of fibrin glue or polyglycolic acid is effective. In cases of massive air leak, pericardial fat and subcutaneous fat are often reported to be effective in covering the leak point [77,78,79,80], and the technique can also be performed under U-VATS. Although there are some limitations in the use of instruments and the surgical view in U-VATS, the control of the needle is easy because the thoracic cavity space is larger than before lung resection. We used to perform intercostal nerve block to the ICS via a transthoracic approach [66]. However, it was reported that a percutaneous approach is more effective than a transthoracic approach in anesthesia [81]. Therefore, we now administer 20 mL of levobupivacaine percutaneously for three intercostal spaces. After the intercostal nerve block, a thoracic drain is placed and the uniport is closed.

## 5. Uniportal VATS Segmentectomy

Our experience with U-VATS segmentectomy is shown in Figure 7. All procedures, including complex segmentectomies, can be performed using U-VATS. Five of these cases are presented as surgical videos.

### 5.1. Right Lower Lobe Apical Segmentectomy (S^6^)

In this case, the tumor was located caudal to S6 and proximal to V6bc. Preoperative CT-guided dye marking was performed to ensure the tumor margin in the intersegmental plane with S10 (Appendix A).

(I)The uniport is placed on the lateral side of the 6th ICS.(II)The PA sheath is dissected, and #11s LND is performed.(III)A6 is cut.(IV)V6 is dissected from the dorsal side.(V)#12L LND is performed, and B6 is cut.(VI)A6 and B6 stumps are ligated together.(VII)The intersegmental planes are identified using ICG.(VIII)Referring to the preoperative marking, stapling is performed between S6 and the basal segment with a sufficient tumor margin.

### 5.2. Left Upper Lobe Tri-Segmentectomy (S^1+2+3^) with Fissureless Technique

A fissureless technique was required because this patient had an incomplete fissure (Appendix A).

(I)The uniport is placed on the anterior side of the 5th ICS.(II)The pleura is incised from the hilum to expose V1-3 and cut.(III)The distal stump of the PV is pulled posteriorly, and the branches of A3 are dissected. The PA branches are cut.(IV)The mediastinal lingual PA is exposed to the periphery, and the stump of the PV and lung is dissected.(V)A1 + 2ab and A1 + 2c are cut.(VI)#12U LND is performed, and B1-3 are exposed. B1-3 are cut with attention to the posterior PA.(VII)The PV stump is ligated.(VIII)The intersegmental planes are identified using ICG.(IX)Stapling is performed between the superior and lingual segments in a straight line from the anterior to the posterior.

### 5.3. Right Upper Lobe Horizontal and Lateral Sub-Segmentectomy (S^2b+S3a^)

This case represents a complex subsegmentectomy for an early-stage NSCLC of less than 2 cm in size (Appendix A).

(I)The uniport is placed on the lateral side of the 6th ICS.(II)Wedge resection is performed. The tumor was diagnosed as adenocarcinoma.(III)The upper-lower fissure is separated, and LN#11s LND is performed (no metastasis).(IV)The anterior type II PV (V2t + V2c + V2b) is dissected and cut.(V)The distal stump of the PV is pulled cranially, and B2b and B3a are exposed.(VI)B2b and B3a are cut.(VII)A3a and A2b are dissected and cut on the posterior side of the bronchial stumps.(VIII)The intersegmental planes are identified using ICG.(IX)All stumps are ligated, and stapling between the intersubsegmental planes is performed under thread control.

### 5.4. Left Lower Lobe Dorsobasal Segmentectomy (S^10^) with Fissure-Based Approach

This case involved left S10 segmentectomy using the interlobar approach (Appendix A).

(I)The uniport is placed on the anterior side of the 6th ICS.(II)The fissure is separated, and the basal PA is identified.(III)V6 is dissected.(IV)This case has a narrow space between V6bc and A10. Therefore, A10 is cut first.(V)Tape is used to tunnel between S6 and S10. The intersegments are then separated.(VI)B10 is dissected and cut.(VII)V10 is identified and cut.(VIII)The intersegmental planes are identified using ICG.(IX)All stumps are ligated, and stapling between the intersegmental planes is performed under thread control.

### 5.5. Left Upper Lobe Apicodorsal Segmentectomy (S^1+2^) and Lower Lobe Ventro-Laterobasal Segmentectomy (S^8+9^)

In this case, two complex segmentectomies were performed simultaneously in U-VATS (Appendix A).

(I)The uniport is placed on the anterior side of the 6th ICS.(II)The fissure is separated, and the interlobar PAs are widely exposed.(III)The branches of A1+2 are cut.(IV)B1 + 2c and B1 + 2 ab are exposed and cut.(V)V1 + 2b behind B1 + 2 is identified and cut.(VI)The intersegmental planes are identified using ICG, stapling between S1 + 2 and S3 is performed, and the left upper lobe apicodorsal segmentectomy is completed.(VII)Next, the interlobar PA is exposed to the periphery, and A5, A8a, and A8b are identified.(VIII)This case had a common trunk of the PV; therefore, the anterior interlobar space was narrow.(IX)A8a is cut first, and the interlobe is separated next.(X)A8b and A9 are cut.(XI)B8+9 is dissected and cut.(XII)V8+9 is dissected and cut.(XIII)The identification of the intersegmental plane using ICG is performed again.(XIV)All stumps are ligated and stapling between the intersegmental planes is performed under thread control.

## 6. Limitations

This study is associated with several limitations. First of all, this article is a narrative review rather than a systematic review. It describes basic intercostal U-VATS segmentectomy and does not mention the subxiphoid approach U-VATS or robot-assisted reduced port surgery.

## 7. Conclusions

The potential of U-VATS has been developed and proven to be beneficial through the efforts of various uniport surgeons. We believe that further development will continue worldwide and that it will be widely recognized as one of the best minimally invasive procedures for early-stage NSCLC. Videos of other surgeries not featured in this special issue are available on the authors’ YouTube channel (Nabetaku Channel: https://www.youtube.com/@taku527 (accessed on 15 May 2024). We hope that these findings will be helpful in learning surgical techniques for U-VATS.

## Figures and Tables

**Figure 3 cancers-16-02343-f003:**
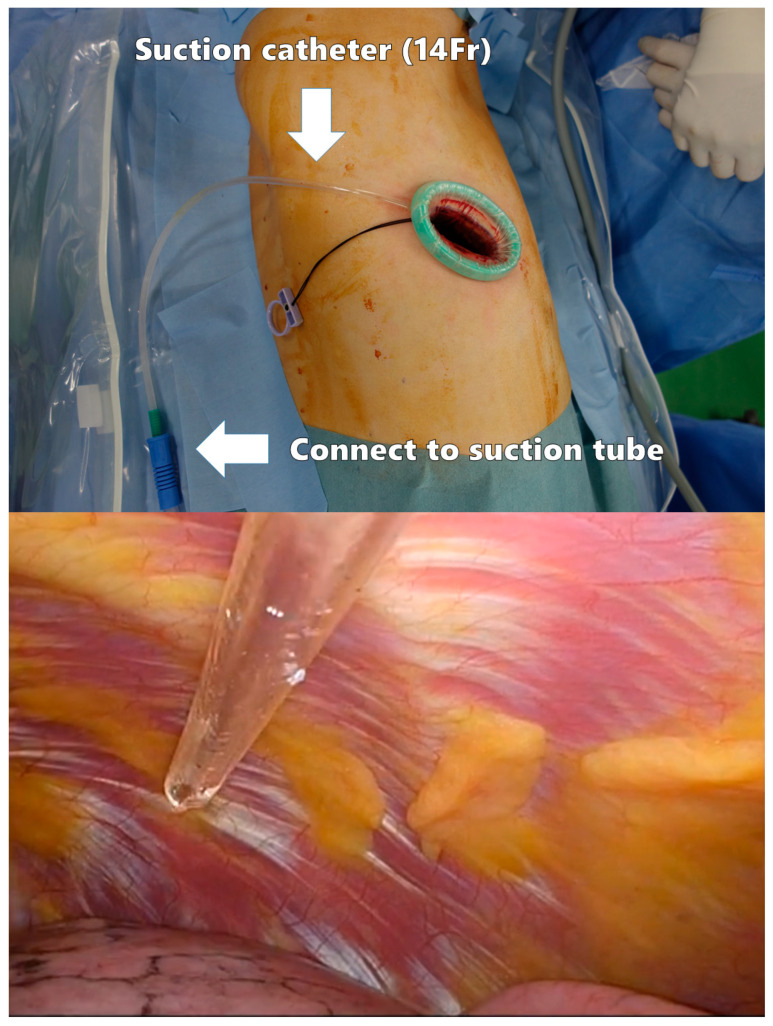
Smoke evacuation device in the surgical field.

**Figure 4 cancers-16-02343-f004:**
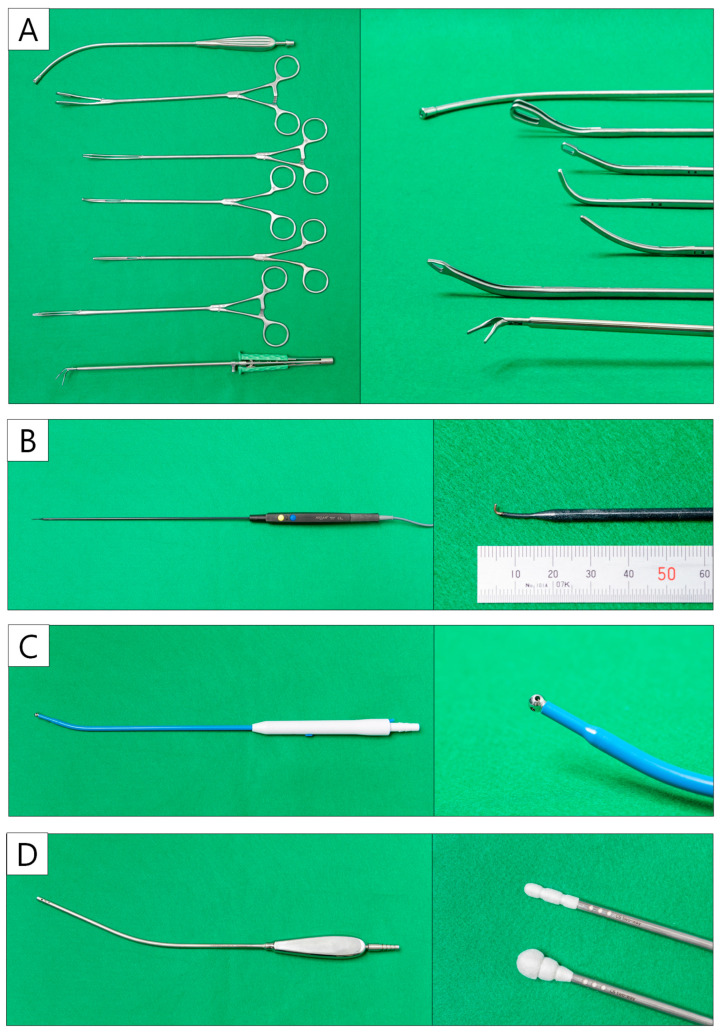
Uniportal VATS instruments. (**A**) The curved forceps and suction device. (**B**) Hook-type electrocautery. (**C**) Curved suction soft coagulator. (**D**) CS two-way handle.

**Figure 5 cancers-16-02343-f005:**
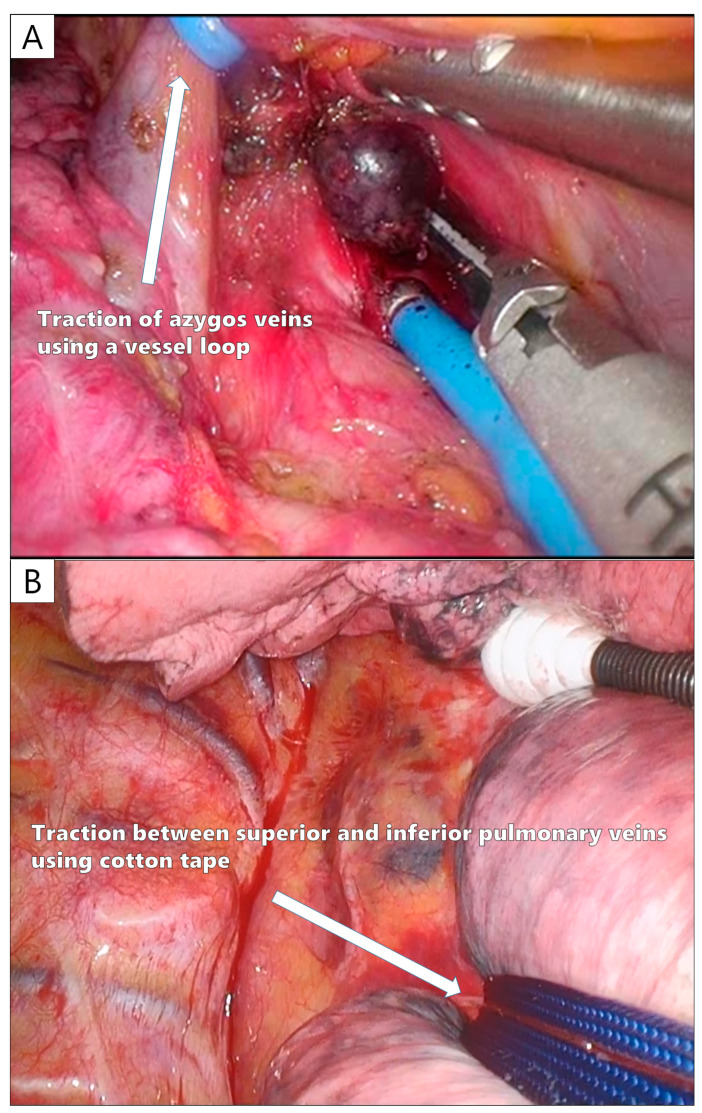
The tips of a uniportal VATS lymph node dissection. (**A**) Taping and control of the azygos vein during upper mediastinal lymph node dissection. (**B**) Taping between the superior and inferior pulmonary veins during lower mediastinal lymph node dissection.

**Figure 6 cancers-16-02343-f006:**
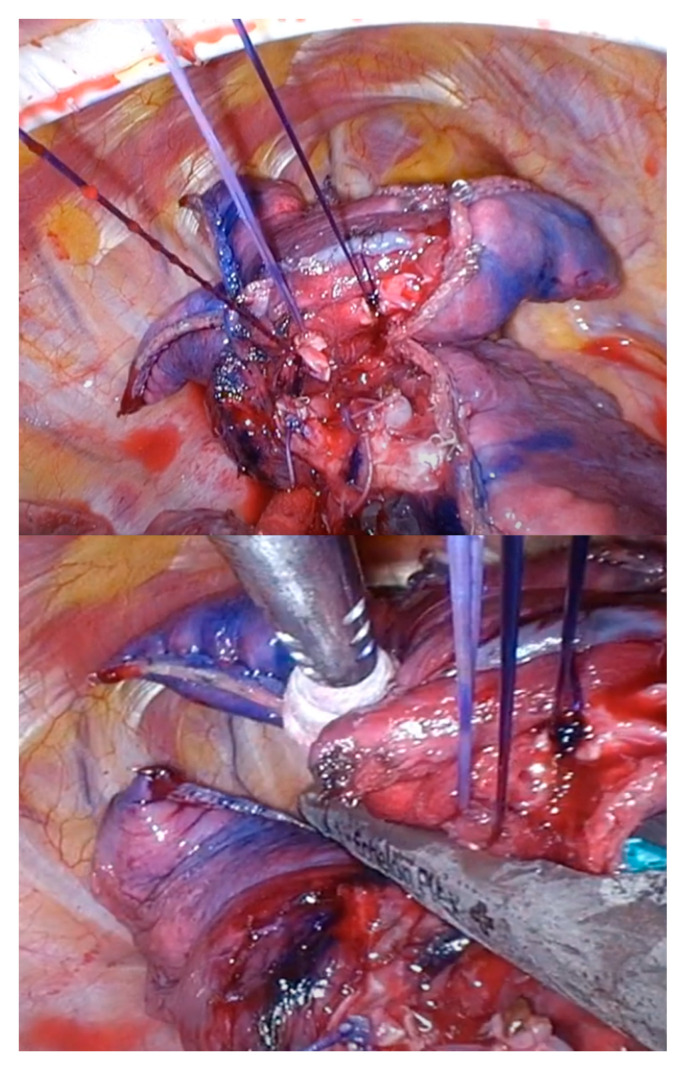
Thread control during intersegmental plane stapling.

**Figure 7 cancers-16-02343-f007:**
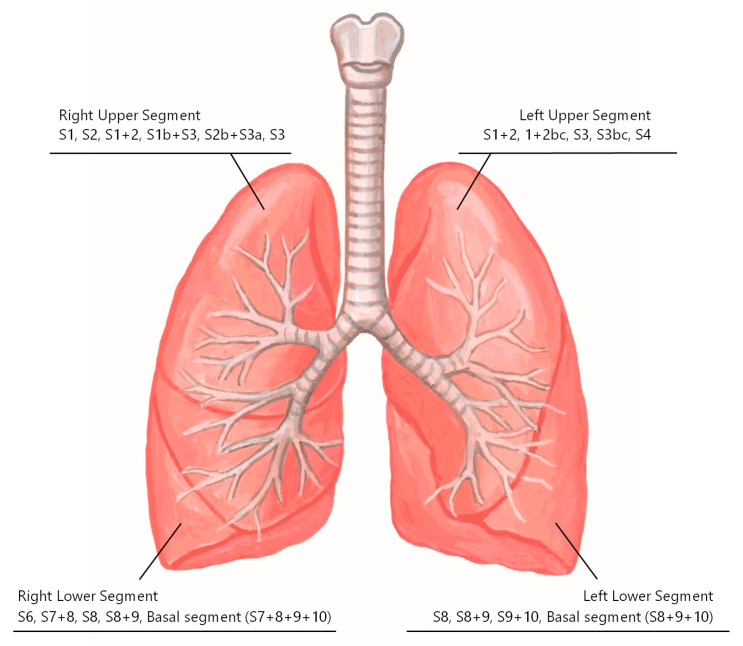
Our experience with uniportal VATS segmentectomy.

**Table 1 cancers-16-02343-t001:** Studies comparing uniportal VATS and multiportal VATS.

Study	Type	Patients (*n*)	U-VATS (*n*)	M-VATS (*n*)	Procedure	Operation Time (min)			Blood Loss (mL)			Conversion to Thoracotomy (%)			Number of Dissected LNs			Drainage Duration (Days)			Postoperative Hospital Stay (Days)			Complication Rate (%)			30-Day Mortality (%)		
						U-VATS	M-VATS	*p* Value	U-VATS	M-VATS	*p* Value	U-VATS	M-VATS	*p* Value	U-VATS	M-VATS	*p* Value	U-VATS	M-VATS	*p* Value	U-VATS	M-VATS	*p* Value	U-VATS	M-VATS	*p* Value	U-VATS	M-VATS	*p* Value
Ruan 2024 [11]	Retrospective (PSM)	322	134	1188	Lobectomy	160.83 ± 71.62	180.67 ± 87.56	0.05	50.37 ± 46.13	85.18 ± 115.97	<0.001	NA			10.79 ± 7.18	10.37 ± 6.78	0.754	4.30 ± 2.75	4.23 ± 2.83	0.841	6.35 ± 2.74	6.96 ± 2.63	0.011	6.7	5.9	0.751	0	0	NA
Zhong 2021 [12]	Retrospective (PSM)	172	93	79	Lobectomy	NA			100 (80–240)	120 (90–260)	0.045	NA			NA			4 (3–6)	5 (3–7)	0.247	9 (7–17)	10 (7–21)	0.354	12.7	15.5	0.354	0	0	NA
Bourdages-Pageau 2019 [13]	Retrospective (PSM)	722	274	448	Lobectomy	137 ± 45	162 ± 49	<0.001	50 (25–100)	100 (50–150)	<0.001	4.0	7.0	0.18	NA			4.5 ± 4.0	5.8 ± 3.5	0.001	4.5 ± 6.5	5.2 ± 5.5	<0.001	NA			0	1	0.498
Hirai 2019 [14]	Retrospective	212	70	142	Lobectomy	165 ± 19	152 ± 18	0.15	65 ± 11	72 ± 12	0.44	5.7	7.0	0.48	20 ± 5	20 ± 5	0.47	1.8 ± 0.9	1.9 ± 0.8	0.54	8.4 ± 1.2	7.8 ± 1.4	0.09	8.6	8.4	0.67	0	0	NA
Wu 2019 [15]	Retrospective	453	197	256	Lobectomy	139.05 ± 81.32	127.27 ± 161.22	0.34	136.12 ± 212.13	178.61 ± 173.17	0.002	5.1	4.3	0.88	15.58 ± 2.12	16.04 ± 9.90	0.52	4.31 ± 3.12	6.93 ± 3.10	<0.001	5.49 ± 4.77	7.23 ± 4.24	<0.001	16.8	10.9	0.09	NA		
Wang 2015 [16]	Retrospective (PSM)	233	50	183	Lobectomy & segmentectomy	169.9 ± 39.58	191.2 ± 51.82	0.029	53.04 ± 47.09	95.33 ± 107.00	0.017	NA			27.39 ± 12.28	22.07 ± 11.18	0.032	NA			5.83 ± 1.83	6.50 ± 2.38	0.132	8.7	17.39	0.216	0	0	NA
Liu 2016 [17]	Retrospective	442	100	342	Lobectomy	179.4 ± 52.2	208.2 ± 63.6	<0.001	55.68 ± 52.81	78.28 ± 84.99	0.001	NA			28.47 ± 11.77	25.23 ± 11.30	0.013	NA			5.96 ± 1.69	6.80 ± 3.56	0.001	8.0	13.7	0.167	NA		
Zhou 2023 [18]	Retrospective (PSM)	2630	400	2230	Segmentectomy	106.95 ± 32.20	98.47 ± 38.09	<0.001	30.98 ± 11.88	43.78 ± 8.51	<0.001	0	0.5	0.35	6.38 ± 2.97	6.70 ± 3.53	0.11	NA			4.25 ± 2.03	4.17 ± 2.29	0.14	4.25	4.67	0.84	NA		
Numajiri 2022 [19]	Retrospective (PSM)	180	57	123	Segmentectomy	141 ± 46	174 ± 45	<0.001	41 ± 83	28 ± 45	0.288	5.3	1.8	0.618	NA			1.5 ± 1.2	2.3 ± 1.8	0.007	3.4 ± 2.0	4.6 ± 2.5	0.006	8.8	5.3	0.716	NA		
Liu 2016 [17]	Retrospective	96	49	47	Segmentectomy	200.4 ± 55.8	207.0 ± 55.2	0.542	63.88 ± 79.60	59.36 ± 50.23	0.739	NA			19.47 ± 10.79	17.91 ± 10.28	0.472	NA			5.76 ± 1.98	6.83 ± 2.21	0.014	6.1	17.0	0.117	NA		

Abbreviations: U-VATS: uniportal video-assisted thoracoscopic surgery, M-VATS: multiportal video-assisted thoracoscopic surgery, LNs: lymph nodes, PSM: propensity score matching, NA: not applicable. Footnote: Continuous variables are presented as the mean ± standard deviation or median (range).

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
