# Peer review of "Uniportal Video-Assisted Thoracoscopic Segmentectomy for Early-Stage Non-Small Cell Lung Cancer: Overview, Indications, and Techniques"

_cancers, 2024, doi:10.3390/cancers16132343_

Round 1
Reviewer 1 Report
Comments and Suggestions for Authors
Comments to authors:
The authors reviews previous research on uniportal video-assisted thoracoscopic segmentectomy, including indications, instrument selection, tumor marking, intersegmental plane identification, lymph node dissection, and surgical videos.
The topic is interesting.
I have the following concerns.
Comment 1
In the material and methods section, the formula used (with descriptors) should be included in the search for the articles included in this review.
Comment 2
Please clarify that the review style for this paper is a narrative review.
Comment 3
Discussion and limitation should be described in this paper.
Comment 4
Although each of the surgical procedures is described in detail, the surgical procedures are general and not specific. It is not necessary to describe all of these procedures. It would be acceptable to supplement the comments on the surgical procedures within the video if necessary.
Comment 5
Although papers on lobectomy are cited in Table 1, these can be omitted.
Instead, data on the rate of complex segmental resection, cancer recurrence, and OS should be added if available.
Comment 6
Certainly, there are few papers comparing uniportal and multiportal approaches, but the authors need to enhance the table in accordance with the theme of the paper.
In the table, it is not necessary to only list papers that compare multiportal VATS and uniportal VATS segmentectomy. As a supplement, please add any volume data on uniportal segmentectomy that would be useful without a comparison to multiportal VATS.
Comment 7
One recent topic is the single port approach with robotic-assisted surgery.
I recommend that this topic be included in the discussion.
Comment 8
Is this approach really effective for lung cancer surgery, the key is cancer recurrence and OS. Supplementation of these results is desired.
Author Response
The authors reviews previous research on uniportal video-assisted thoracoscopic segmentectomy, including indications, instrument selection, tumor marking, intersegmental plane identification, lymph node dissection, and surgical videos.
The topic is interesting.
I have the following concerns.
Comment 1
In the material and methods section, the formula used (with descriptors) should be included in the search for the articles included in this review.
Thank you for your important comments. In writing this review manuscript, I have searched the articles using a very large number of keywords. It is impossible to enumerate them all. Also, in the Cancers previous review articles that was presented to me by the editor for reference, there was no mention of how to search for articles. Thank you very much for your comments.
Comment 2
Please clarify that the review style for this paper is a narrative review.
Thank you for your wonderful comments. I have clarified at the end of the introduction that this is a narrative review (line 49).
Comment 3
Discussion and limitation should be described in this paper.
Thanks for your comment, I have looked at previous review articles on Cancers and found no chapters of discussion or limitation. This is a review article and I believe each section will include content similar to discussion.
Comment 4
Although each of the surgical procedures is described in detail, the surgical procedures are general and not specific. It is not necessary to describe all of these procedures. It would be acceptable to supplement the comments on the surgical procedures within the video if necessary.
Thank you for your comment. It was difficult to explain in the surgical videos due to the limited data capacity. Also, I want beginners to U-VATS to read this article, so I have explained it in detail. Thank you.
Comment 5
Although papers on lobectomy are cited in Table 1, these can be omitted.
Thank you for your comment. In this section, I focused on the comparison between U-VATS and M-VATS, and I thought it was not necessary to limit it to segmentectomy. Thank you.
Instead, data on the rate of complex segmental resection, cancer recurrence, and OS should be added if available.
As you say, this is very important. There were only a few studies that mentioned prognosis, so we did not include it in the table, but we did mention it in the text (lines 64-71). Thank you very much.
Comment 6
Certainly, there are few papers comparing uniportal and multiportal approaches, but the authors need to enhance the table in accordance with the theme of the paper.
In the table, it is not necessary to only list papers that compare multiportal VATS and uniportal VATS segmentectomy. As a supplement, please add any volume data on uniportal segmentectomy that would be useful without a comparison to multiportal VATS.
Thank you for your comment. In fact, there are many papers comparing U-VATS and M-VATS. On the other hand, there are few reports on U-VATS segmentectomy (most of which are related to the learning curve as mentioned in 3.1). Therefore, I have collected the few reports in this review manuscript and summarized them in each section. Thank you very much.
Comment 7
One recent topic is the single port approach with robotic-assisted surgery.
I recommend that this topic be included in the discussion.
Thank you for your important comment. I have added the phrase "including robotic-assisted reduced port surgery" to the conclusion (line 353-354).
Comment 8
Is this approach really effective for lung cancer surgery, the key is cancer recurrence and OS. Supplementation of these results is desired.
Thank you for your important comments. At present, no disadvantages have been reported for U-VATS in terms of recurrence or OS. I am collecting and analyzing national data in my country. Therefore, I think that I will be able to report large data on long-term prognosis in the future. Thank you for all your comments!

Reviewer 2 Report
Comments and Suggestions for Authors
Authors wrote a nice review about uniportal vats pulmonary segmentectomy.
The paper does not have a particular originality, because most of the issues are already well-known in thoracic surgical community.
However, it may be a good review in a cancer journal
Author Response
Authors wrote a nice review about uniportal vats pulmonary segmentectomy.
The paper does not have a particular originality, because most of the issues are already well-known in thoracic surgical community.
However, it may be a good review in a cancer journal
Thank you for your comments. Although much of the content may already be known, I have explained each section with numerical support. I also wrote this article so that readers can fully understand U-VATS segmentectomy by reading only this article. I think this article is more comprehensive and richer in content than previous articles. Thank you so much!
Reviewer 3 Report
Comments and Suggestions for Authors
Dear Author,
I am delighted to review your article, which gave me a fresh perspective on the process and advantages of U-VATS. However, I have noticed the following issues while reading your article:
- Can U-VATS only be used for early-stage lung cancer? What are the limitations of its application?
- Are there better innovations in U-VATS, rather than just comparing it with thoracotomy?
Thank you for your article.
Author Response
I am delighted to review your article, which gave me a fresh perspective on the process and advantages of U-VATS. However, I have noticed the following issues while reading your article:
Can U-VATS only be used for early-stage lung cancer? What are the limitations of its application?
Are there better innovations in U-VATS, rather than just comparing it with thoracotomy?
Thank you for your article.
Thank you for your important comment. U-VATS can also be applied to advanced lung cancer. However, this time, in accordance with the special feature "Treatment of Early-Stage Non-small Cell Lung Cancer", we focused on U-VATS and segmentectomy as minimally invasive treatments for early-stage lung cancer.
From the viewpoint of making surgical wound smaller, U-VATS is the most minimally invasive surgery. We surgeons cannot make wound zero, we need at least one. I think it is an innovative approach developed with the patient in mind. Thank you very much!
Reviewer 4 Report
Comments and Suggestions for Authors
I would like to congratulate the authors on their interesting article on segmentectomy using the uniportal VATS approach. In this article, the authors review the literature on these topics and present the surgical techniques used in their center. The content is supplemented with high-quality visual materials.
The article is written in very good English. The issues raised by the authors are currently very up-to-date and widely discussed in the thoracic surgical community. The excellent presentation of the surgical technique may be very useful for surgeons involved in the minimally invasive treatment of lung cancer.
I have no comments on the article and suggest its publication in the journal "Cancers".
Comments on the Quality of English LanguageVery good English quality.
Author Response
I would like to congratulate the authors on their interesting article on segmentectomy using the uniportal VATS approach. In this article, the authors review the literature on these topics and present the surgical techniques used in their center. The content is supplemented with high-quality visual materials.
The article is written in very good English. The issues raised by the authors are currently very up-to-date and widely discussed in the thoracic surgical community. The excellent presentation of the surgical technique may be very useful for surgeons involved in the minimally invasive treatment of lung cancer.
I have no comments on the article and suggest its publication in the journal "Cancers".
Great comments, thank you! I'm glad to hear you say that, and happy to have written this invited manuscript. Thank you very much!
Round 2
Reviewer 1 Report
Comments and Suggestions for Authors
The author responded accurately to my questions.
Author Response
3.3.1. Locating the Tumor section, should comment on those tumours that require marking, for example deep ones or those near intersegmental planes etc..
Thank you for your great comment. This is very important, but I forgot mention it. I added the following paragraph about it in Line 129-134 (3.3.1. Locating the Tumor). Thank you very much!!
“We perform preoperative marking in cases where the tumor may not be visible from the pleura and where the lesion mainly demonstrates a ground-glass opacity. In addition, we consider that cases in which the tumor is located somewhat deep in the lung and thus may move in the parenchyma are not good indications for U-VATS segmentectomy, and such cases should instead be dissected between the intersegmental planes with palpation under thoracotomy.”
Should comment on limitation of this review covering only some aspects of uVATS segmentectomy and primarily authors’ techniques. Others including subxiphoid uVATS segmentectomy have been described and reported and not covered here in this article , as the authors’ focus is intercostal uVATS.
Thank you for your great comment. Some of the previous cancer review articles did not include limitations, but based on your comments, we decided to include limitations in this article. We have added the following in the limitations section (Line 362-366), which includes the points you raised. Thank you very much!!
“This study is associated with several limitations. First of all, this article is a narrative re-view rather than a systematic review. It describes basic intercostal U-VATS segmentectomy and does not mention the subxiphoid approach U-VATS or robot-assisted reduced port surgery.”
They should also provide some insight for the reader of the advantages and disadvantages to compare their technique of UVATS segmentectomy with other techniques described. For example, they mention they stand right of patient so use right hand for smoother dissection, but do not mention the potential disadvantage of having to learning the anterior and posterior hilar approach.
How about where the assistant who holds the scope is positioned? How does that compare with what is described in literature? What are pros and cons?
Thank you for your great comment. A disadvantage of this approach is that the intercostal space is narrower in the mid-axillary line than in the anterior axillary line. However, because the thoracic cavity can be viewed from above and the operation is intuitive, there is no need to learn anterior or posterior approaches (since the operation can be performed with the same field of view as in thoracotomy).
In response to your comment, we have added the following to 4.2. Approach and Incision (Line 169-170, and Line 171-176). Thank you very much!!
“The first assistant and scopist stand on the opposite side of the main surgeon.”
“The strength of this approach is that it provides a bird's-eye view of the thoracic cavity. It is therefore possible to secure a view of not only the hilum, but also the dorsal side, thus making it suitable for dorsal pleural dissection and lymph node dissection in the upper and lower mediastinal regions. However, the ICS is slightly narrower at the mid-axillary line than at the anterior axillary line, which may be a disadvantage in small patients with a narrow ICS.”
Thank you for your great comments.
Your help has made my paper much better. I am very grateful. Thank you very much.